# Valorization of Acid Leaching Post-Consumer Gypsum Purification Wastewater

Miguel Castro-Díaz [1,*], Sergio Cavalaro [1], Mohamed Osmani [1], Saeed Morsali [1], Matyas Gutai [1], Paul Needham [2], Bill Parker [3] and Tatiana Lovato [3]

[1] School of Architecture, Building and Civil Engineering, Loughborough University, Loughborough LE11 3TU, UK; s.cavalaro@lboro.ac.uk (S.C.); m.osmani@lboro.ac.uk (M.O.); s.s.morsali@lboro.ac.uk (S.M.); m.gutai@lboro.ac.uk (M.G.)
[2] ENVA, Enviro Building, Private Road 4, Colwick Industrial Estate, Nottingham NG4 2JT, UK; paul.needham@enva.com
[3] British Gypsum, East Leake, Loughborough LE12 6JT, UK; bill.parker@saint-gobain.com (B.P.); tatiana.lovato@saint-gobain.com (T.L.)
[*] Correspondence: m.castro-diaz@lboro.ac.uk

**Abstract:** Industries are required to utilize treatment technologies to reduce contaminants in wastewater prior to discharge and to valorize by-products to increase sustainability and competitiveness. Most acid leaching gypsum purification studies have obviated the treatment of the highly acidic wastewater produced. In this work, acidic wastewater from acid leaching purification of post-consumer gypsum was treated to recover a valuable solid product and reusable water. The main aims of this work were to determine the impact of recirculating acidic and treated wastewaters on the efficiency of the acid leaching purification process and to valorize the impurities in the wastewater. Samples were characterized through X-ray fluorescence and X-ray diffraction. SimaPro 9.5 and the ReCiPe 2016 midpoint method were used for the life cycle assessment of three sustainable wastewater management approaches. The reuse of the acidic wastewater did not improve the chemical purity of gypsum. Soluble impurities were precipitated at pH 10.5 as a magnesium-rich gypsum that could be commercialized as fertilizer or soil ameliorant. The alkaline-treated water was reused for six acid leaching purification cycles without impacting the efficiency of the purification process. An acid leaching–neutralization–filtration–precipitation approach demonstrated superior overall environmental performance. Barriers and enabling measures for the implementation of an in-house wastewater treatment were identified.

**Keywords:** refurbishment plasterboard waste; demolition plasterboard waste; gypsum waste recycling; acid leaching purification; wastewater treatment; wastewater valorization

## 1. Introduction

Post-consumer plasterboard wastes from refurbishment and demolition projects are currently recycled through physical processes that rely on manual segregation, crushing, sieving and magnetic separation. However, these plasterboard wastes contain water-soluble impurities, such as chloride, magnesium, sodium, and potassium salts, that migrate to the paper–gypsum core interface during plasterboard drying and affect paper bonding during plasterboard production [1]. These water-soluble impurities cannot be removed with current plasterboard recycling processes; thus, chemical treatments are necessary to produce the high-purity recycled gypsum demanded by plasterboard manufacturers [1]. The combination of a modified mechanical treatment and an acid leaching purification process was proven to be an effective technology to produce high-purity (>96 wt%) post-consumer gypsum [2]. Acidic wastewater with dissolved impurities was obtained as a by-product of the acid leaching purification process. This acidic wastewater is classified as hazardous because it is highly corrosive and the impurities can contribute to eutrophication [3]. In the

European Union, the release of industrial wastewater is regulated by the Water Framework Directive [4], whereas in the United States, the Environmental Protection Agency (EPA) has set strict limits for the maximum contaminant levels allowed to be present in industrial wastewater and discharge water through the Clean Water Act [5]. Therefore, industries are required to utilize water treatment technologies that reduce contaminants in wastewater to within acceptable limits prior to discharge.

The acidic wastewater generated in the acid leaching gypsum purification process could be reused or treated to minimize disposal costs. For instance, Chen et al. [6] reused a sulfuric acid ($H_2SO_4$) solution five times without impacting the purity of the acid-leached gypsum product. Alternatively, calcium hydroxide, $Ca(OH)_2$ could be reacted with residual $H_2SO_4$ in the wastewater to produce gypsum, $CaSO_4 \cdot 2H_2O$, according to the following equation:

$$Ca(OH)_{2(s)} + H_2SO_{4(l)} \rightarrow CaSO_4 \cdot 2H_2O_{(s)} \tag{1}$$

Chen et al. [6] precipitated $CaSO_4 \cdot 2H_2O$ when the pH of the acid leaching solution obtained after five cycles was adjusted to 1.78 with the addition of $Ca(OH)_2$. When the pH was raised to 3.61, the precipitate was $CaSO_4 \cdot 2H_2O$ with amorphous iron hydroxide, $Fe(OH)_2$, and when the pH was raised to 7.95, the precipitate was $CaSO_4 \cdot 2H_2O$ that was yellow in color, indicating the presence of iron. These authors suggested that the treated water could be reused in the preparation of the $H_2SO_4$ solution for the acid leaching purification process. This would align with the EU's Circular Economy Action Plan [7], which aims to reduce, reuse, and recycle waste streams to increase resource sustainability and improve business competitiveness. In this regard, acidic wastewater generated by metal, paper, and leather industries has been treated with neutralizing chemicals to precipitate soluble metal ions [8].

Most acid leaching gypsum purification studies have been performed with phosphogypsum, which is a synthetic gypsum obtained as a by-product of the phosphoric acid industry, to produce a purified material that could be used as set retarder in cement [9–13] or as plaster [14–18]. Other studies have used acid leaching to extract valuable rare earth elements in phosphogypsum [19–25]. Nonetheless, most studies have obviated the valorization of the acidic wastewater obtained from acid leaching gypsum purification. Only the study by Chen et al. [6] with red gypsum from titanium oxide manufacturing evaluated the reuse and treatment of the acidic wastewater at different pH, but no attempt was made to optimize the wastewater treatment and determine its environmental and economic impacts. Therefore, there are currently no comprehensive studies of the valorization of the acidic wastewater obtained from acid leaching gypsum purification.

The main aim of this work was to determine the optimum wastewater treatment conditions to preserve acid leaching post-consumer gypsum purification process efficiency and minimize the environmental and economic impacts of this purification process to increase its sustainability. To this end, three sustainable wastewater management approaches were investigated, the exploitation potential of the best wastewater treatment was assessed, and an in-house wastewater treatment for the acid leaching gypsum purification plant was proposed. The three sustainable wastewater management approaches were: (i) acid leaching–filtration, with reuse of the acidic wastewater in the acid leaching gypsum purification process; (ii) acid leaching–filtration–precipitation, with reuse of the treated water in the acid leaching gypsum purification process; and (iii) acid leaching–neutralization–filtration–precipitation, with reuse of the treated water in the acid leaching gypsum purification process.

## 2. Materials and Methods

### 2.1. Sourcing of Materials

Sourcing of the refurbishment plasterboard waste (RPW) and demolition plasterboard waste (DPW) can be found elsewhere [2]. The preparation of gypsum from RPW (GRPW) and gypsum from DPW (GDPW) was as follows. First, RPW and DPW were manually segregated to separate large contaminants, such as mortar, plastics, paint, and foam strips.

Then, the segregated plasterboards were broken down into small pieces of around 2–5 cm before crushing and sieving. Crushing was carried out with a porcelain mortar and pestle, and sieving was performed with two stainless steel Impact sieves and a receiver pan that were 300 mm in diameter and conform to standards ISO 3310-1 and BS 410-1 [26,27]. The sieves had apertures of 2000 μm and 250 μm and were stacked together with the receiver pan. In the first crushing stage, the small plasterboard pieces were ground carefully to obtain particles less than 2000 μm in size and to separate impurities, such as paper fragments, during sieving. In a second crushing stage, gypsum particles that were 250–2000 μm in size were ground to less than 250 μm to produce GRPW and GDPW feedstocks for acid leaching purification tests. GRPW and GDPW had particle sizes <250 μm and contained <0.5 wt% paper fibers, which were generated during the crushing stages.

$H_2SO_4$ (Fisher Chemicals, Loughborough, UK, certified analytical reagent, minimum purity 95 vol%) and $Ca(OH)_2$ (Acros Organics, Loughborough, UK, ACS reagent, >95 wt%) were used for acid leaching purification tests and wastewater treatments, respectively. Purified or municipal water was used to prepare the 3 wt% and 5 wt% $H_2SO_4$ solutions used in this work.

### 2.2. Experimental Design

Figure 1 presents the steps and main objectives of the three sustainable wastewater management approaches (WMAs) evaluated in this research. WMA 1 considers the reuse of acidic wastewater in the acid leaching purification process. WMA 2 precipitates the soluble impurities in the wastewater and reuses the treated water in the acid leaching gypsum purification process. WMA 3 is similar to WMA 2, with the only difference being that the acidic gypsum slurry obtained after the acid leaching step is neutralized prior to filtration. The goals of these approaches are to (i) reduce water consumption in the acid leaching gypsum purification process (WMA 1, WMA 2, and WMA 3); (ii) valorize impurities in the wastewater (WMA 2 and WMA 3); and (iii) minimize the use of costly corrosion-resistant equipment in the acid leaching purification process (WMA 3).

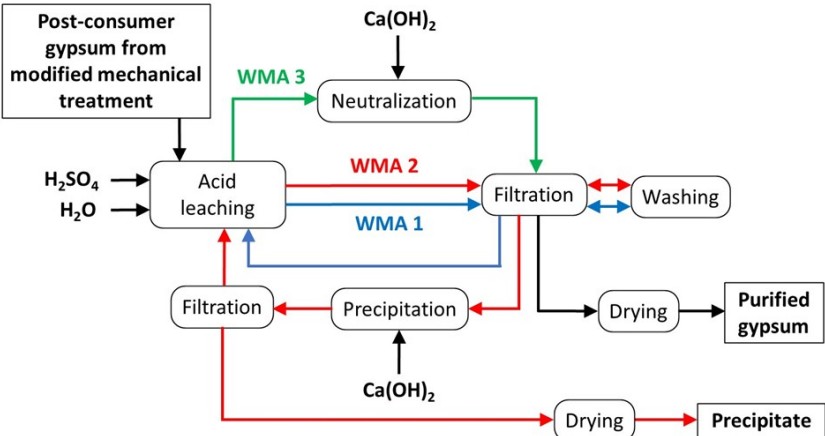

**Figure 1.** Schematic representation of the steps of the three sustainable wastewater management approaches (WMAs) investigated in this work.

### 2.3. Tests

Acid leaching purification tests in WMA 1, WMA 2, and WMA 3 were performed with a borosilicate beaker, a hot plate, and an overhead stirrer placed inside a fume cupboard, in accordance with our previous work [2]. Briefly, either purified or municipal water was used to prepare the acidic solutions. The temperature of the gypsum slurry was measured with an independent thermocouple placed inside the beaker. Acid leaching purification tests were performed at 90 °C for either 1 h using a 5 wt% (0.5 M) $H_2SO_4$ solution or for 2 h using a 3 wt% (0.3 M) $H_2SO_4$ solution, with a gypsum/solution ratio of 1:3 wt/wt, a slurry volume of 350 mL, and a stirring speed of either 50 or 150 revolutions per minute. GRPW

and GDPW were added to the $H_2SO_4$ solution at room temperature, and the gypsum slurries were heated to 90 °C at a rate of 3–4 °C/min. The slurry was cooled down to room temperature at the end of each acid leaching purification test. A Büchner funnel and filtering flask connected to a vacuum pump were used to recover the purified gypsum. Whatman filter paper grade 1 was used with the filtration kit.

Washing of the gypsum cakes in WMA 1 and WMA 2 was performed with purified water, and the washing process was completed when the pH of the solution, which was measured with litmus paper, was around 5.5. The purified gypsum cakes were dried in an electric furnace at 45 °C for 24 h.

In precipitation tests in WMA 2, the acidic wastewater obtained after acid leaching gypsum purification and filtration was placed in a borosilicate beaker with a magnetic stir bar. The beaker was placed on the top plate of a magnetic stirrer and $Ca(OH)_2$ was added slowly to the wastewater at room temperature until reaching a pH of 5.5, 8.0, or 10.5. The pH was measured with a bench-top Hanna Instruments HI-2211 pH meter. In sequential acid leaching and precipitation tests in WMA 2, treated water obtained after precipitation at pH 10.5 was used to prepare 3 wt% $H_2SO_4$ solutions for the next acid leaching purification test.

The acidic gypsum slurry obtained in WMA 3 after acid leaching gypsum purification was neutralized with $Ca(OH)_2$ to pH 5.5 using the same setup and methodology used during precipitation in WMA 2.

### 2.4. X-ray Fluorescence

The chemical composition of the samples was determined through X-ray fluorescence (XRF). XRF analyses were performed with an Orbis micro-XRF spectrometer. Sample pellets were prepared by blending 0.8 g of gypsum powder with 0.2 g of boric acid powder (binder). The blend was compacted using a die and piston of 5 mm in diameter that was placed in a manual hydraulic press. Each XRF pellet was obtained after applying 10 tons of force on the piston. XRF data were acquired under vacuum in five regions of the pellet using a voltage of 30 kV, a current of 0.4 mA, an amplifier time of 1.6 μs, and an acquisition time of 120 s. The weight percentages of $SO_3$, CaO, $SiO_2$, $Al_2O_3$, $Fe_2O_3$, MnO, MgO, $P_2O_5$, $K_2O$, $Na_2O$, $Ni_2O_3$, SrO, and Cl were recorded. The chemical purity values of the gypsum samples were calculated as the sum of the CaO, $SO_3$, $SiO_2$, $Al_2O_3$, and $Fe_2O_3$ contents, and their mean standard deviation values were determined. Furthermore, the $CaSO_4$ content (sum of CaO and $SO_3$ contents) was used to differentiate between samples with similar chemical purity.

### 2.5. X-ray Diffraction

The mineral composition of the samples was determined through X-ray diffraction (XRD). The contents of gypsum, $CaSO_4 \cdot 2H_2O$, portlandite, $Ca(OH)_2$, kieserite, $MgSO_4 \cdot H_2O$, and brucite, $Mg(OH)_2$, in the precipitates obtained from the acidic wastewater were determined through integration of XRD peaks. XRD patterns were obtained using a Bruker D2 Phaser X-ray diffractometer fitted with a 1-dimensional Lynxeye detector and using Ni-filtered Cu Kα radiation run at 30 kV and 10 mA. Patterns were recorded between 10° and 100° 2θ using a step size of 0.02. The XRD patterns were analyzed with DIFFRAC.EVA v3.1 diffraction software. ICDD-PDF numbers 33-0311, 04-0733, 70-2156, and 07-0239 were used for the semi-quantitative and qualitative analyses of gypsum, portlandite, kieserite, and brucite, respectively.

### 2.6. Life Cycle Assessment

An evaluation of the environmental performances of WMA 1, WMA 2, and WMA 3 was conducted using the life cycle assessment (LCA) method in SimaPro 9.5 with the ReCiPe 2016 midpoint method. Characterization analysis and normalization analysis were performed. Characterization analysis involves assigning values to different environmental impacts based on their relative importance and quantifying their magnitude for comparison.

Normalization analysis is a technique that allows for the comparison of impact categories by scaling their values relative to a reference or benchmark, thus providing a standardized basis for assessing their overall importance and contribution to environmental impact [28].

### 3. Results and Discussion

*3.1. WMA 1*

Table 1 presents the chemical composition of gypsum from refurbishment plasterboard waste (GRPW) and gypsum from demolition plasterboard waste (GDPW). Using equation 1, the chemical purity values of GRPW and GDPW were, respectively, 95.9 wt% and 96.0 wt%.

**Table 1.** Chemical composition, expressed as weight percentage, of gypsum from refurbishment plasterboard waste (GRPW) and gypsum from demolition plasterboard waste (GDPW).

| Sample | $SO_3$ | CaO | $SiO_2$ | $Al_2O_3$ | $Fe_2O_3$ | MnO | MgO | $P_2O_5$ | $K_2O$ | $Na_2O$ | Cl | $Ni_2O_3$ | SrO |
|--------|--------|------|---------|-----------|-----------|------|------|----------|--------|---------|------|-----------|------|
| GRPW | 63.7 | 30.6 | 1.0 | 0.4 | 0.2 | 0.3 | 0.1 | 2.0 | 0.2 | 0.5 | 0.8 | <0.1 | 0.1 |
| GDPW | 62.5 | 30.7 | 1.9 | 0.5 | 0.4 | 0.3 | 0.7 | 2.1 | 0.2 | <0.1 | 0.5 | <0.1 | <0.1 |

Initial acid leaching purification tests were conducted with GDPW at 90 °C for 1 h with a 5 wt% $H_2SO_4$ solution, which were identified as optimum conditions for the purification of gypsum from post-consumer plasterboard wastes [2]. Figure 2 presents the chemical purity values of the initial and acid-leached GDPW samples when using fresh and spent 5 wt% $H_2SO_4$ solutions after two cycles. Spent solutions were recovered after acid leaching purification tests, whereby spent solution 1 was obtained in cycle 1 from the fresh solution and spent solution 2 was obtained in cycle 2 from spent solution 1. The use of the fresh 5 wt% $H_2SO_4$ solution increased the chemical purity of GDPW from 96.0 wt% to 96.7 wt% and increased the $CaSO_4$ content from 93.2 wt% to 94.4 wt%. The use of spent solution 1 caused a 0.2 wt% reduction in the chemical purity to 96.5 wt% and a drop of 0.3 wt% in the $CaSO_4$ content to 94.1 wt% compared to the purified sample obtained with the fresh $H_2SO_4$ solution. The chemical purity decreased a further 0.2 wt% to 96.3 wt% and the $CaSO_4$ content decreased a further 0.8 wt% to 93.3 wt% with the use of spent solution 2. The decrease in acid leaching gypsum purification efficiency with each reuse could be rationalized by the progressive increase in soluble salt content in the spent solutions. These results also suggest that reusing spent $H_2SO_4$ solutions would not be technically viable because the chemical purity and $CaSO_4$ content of the purified gypsum were similar to the chemical purity and $CaSO_4$ content of the GDPW feedstock (96.4 wt% and 94.4 wt%, respectively).

*3.2. WMA 2*

Acidic wastewater produced after one acid leaching purification cycle of GRPW at 90 °C for 1 h using a 5 wt% $H_2SO_4$ solution prepared with municipal water and a stirring rate of 50 rpm was treated with $Ca(OH)_2$ to pH 5.5, pH 8.0, and pH 10.5. The precipitates were filtered and characterized through XRF (Figure 3). At pH 5.5, the precipitate was mainly composed of $CaSO_4$ (63 wt%) and $Fe_2O_3$ (20 wt%). At pH 8.0, the amount of $CaSO_4$ in the precipitate increased to 75 wt% and the amount of $Fe_2O_3$ decreased to 10 wt%. The increase in pH from 5.5 to 8.0 also decreased the $SiO_2$ content from 10 wt% to 5 wt% and increased the MgO content from 1 wt% to 4 wt%. At pH 10.5, the precipitate mainly constituted MgO (47 wt%) and $CaSO_4$ (38 wt%).

In order to obtain a better understanding of the required pH for the complete precipitation of MgO, the precipitates obtained at pH 5.5, 8.0, and 10.5 were characterized through XRD. Table 2 shows that the gypsum content of the precipitate was fairly similar when the pH was 5.5 and 8.0 (94.1–95.8 wt%) but decreased to 86.3 wt% when the pH was 10.5. The reduction in gypsum content with the increase in the pH value was accompanied by an increase in brucite, kieserite, and portlandite. Brucite and kieserite were present in significant amounts at pH 10.5 (3.8 wt% and 3.5 wt%, respectively).

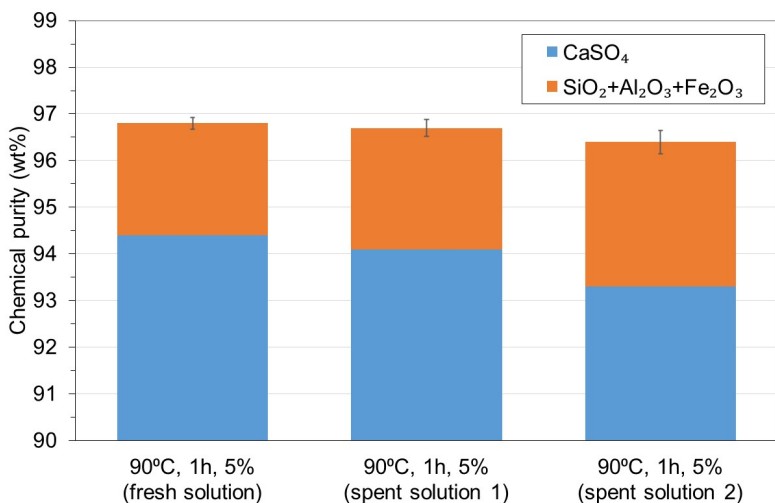

**Figure 2.** Chemical purity of gypsum from demolition plasterboard waste (GDPW) before and after acid leaching purification at 90 °C for 1 h using fresh and spent 5 wt% $H_2SO_4$ solutions and a stirring rate of 150 rpm.

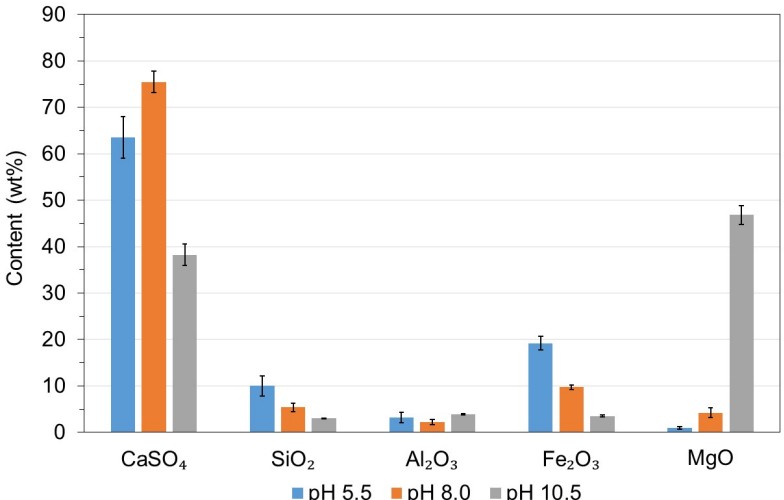

**Figure 3.** Precipitate composition as a function of the final pH after neutralization of acidic wastewater obtained after one acid leaching purification cycle using fresh municipal water.

**Table 2.** Mineral composition expressed as weight percentage of precipitates produced at pH 5.5, 8.0, and 10.5 from acidic wastewater obtained after acid leaching purification of gypsum from refurbishment plasterboard waste (GRPW) at 90 °C for 1 h using a 5 wt% $H_2SO_4$ solution prepared with municipal water and a stirring rate of 50 rpm, and precipitates produced at pH 10.5 from acidic wastewater obtained after acid leaching purification of gypsum from demolition plasterboard waste (GDPW) at 90 °C for 2 h using a 3 wt% $H_2SO_4$ solution prepared with purified water, treated water 1, and treated water 2 and a stirring rate of 50 rpm.

| Precipitates | Gypsum, $CaSO_4 \cdot 2H_2O$ | Portlandite, $Ca(OH)_2$ | Kieserite, $MgSO_4 \cdot H_2O$ | Brucite, $Mg(OH)_2$ |
|---|---|---|---|---|
| GRPW, municipal water, pH 5.5 | 95.8 | 2.2 | 0.8 | 1.2 |
| GRPW, municipal water, pH 8.0 | 94.1 | 3.4 | 0.9 | 1.6 |
| GRPW, municipal water, pH 10.5 | 86.3 | 6.4 | 3.5 | 3.8 |
| GDPW, purified water, pH 10.5 | 79.0 | 4.0 | 9.6 | 7.4 |
| GDPW, treated water 1, pH 10.5 | 81.7 | 3.3 | 6.1 | 8.9 |
| GDPW, treated water 2, pH 10.5 | 87.7 | 1.0 | 6.1 | 5.2 |

In previous studies [29–31], $MgSO_4$ was reacted with $Ca(OH)_2$ to precipitate $Mg(OH)_2$ and $CaSO_4 \cdot 2H_2O$ according to the following equation:

$$MgSO_{4(aq)} + Ca(OH)_{2(s)} + 2H_2O_{(l)} \rightarrow Mg(OH)_{2(s)} + CaSO_4 \cdot 2H_2O_{(s)} \tag{2}$$

The solubility of $Mg(OH)_2$ was found to be significantly lower than that of $CaSO_4$. However, all of these studies have disagreed regarding the required pH to achieve the complete precipitation of $Mg(OH)_2$. For instance, Semerjian and Ayoub [29] found that Mg started to precipitate as $Mg(OH)_2$ at approximately pH 9.5, and significant precipitation occurred at pH 10.5. On the other hand, Xiong et al. [30] indicated that the precipitation rate began to increase with pH ranging from 8.0 to 10.5, with most of the Mg precipitated at pH 10.0. These authors also suggested that $Mg(OH)_2$ precipitation could be accompanied by co-precipitation of iron hydroxides, as found here at pH 5.5 (Figure 3). However, Tolonen et al. [31] found that pH 9.6 was too low for $Mg(OH)_2$ precipitation because 43 wt% of Mg was present as soluble $MgSO_4$ at pH 9.6. On the other hand, all Mg was precipitated as $MgSO_4$ at pH 12.5. The fact that $MgSO_4$ was found to be highly soluble in water at pH 9.6 might suggest that $MgSO_4 \cdot H_2O$ molecules could be trapped inside $Mg(OH)_2$ crystals formed at pH 10.5.

A comparison of the chemical composition of the precipitates obtained with $Ca(OH)_2$ at pH 10.5 from the acidic wastewater generated after acid leaching purification of GRPW and GDPW at 90 °C for 1 h using a 5 wt% $H_2SO_4$ solution is presented in Figure 4. The precipitate from GRPW contained similar $CaSO_4$ content to the precipitate from GDPW (38.2 wt% and 40.3 wt%, respectively). However, the MgO content in the precipitate from GDPW was around 4.5 wt% higher than that in the precipitate from GRPW. The $Na_2O$ content in the precipitate from GDPW (4.8 wt%) was twice as high as that in the precipitate from GRPW (2.4 wt%). On the other hand, the precipitate from GRPW contained between 3 and 4 wt% of $SiO_2$, $Al_2O_3$, and $Fe_2O_3$, representing around 10.5 wt% of the precipitate, whereas the contents of these compounds were below 1 wt% in the precipitate from GDPW. As a result, the combined $CaSO_4$ and MgO contents in the precipitates from GRPW and GDPW were, respectively, around 85 wt% and 92 wt%. The ratio of MgO content to $CaSO_4$ content in both precipitates was close to 1.25.

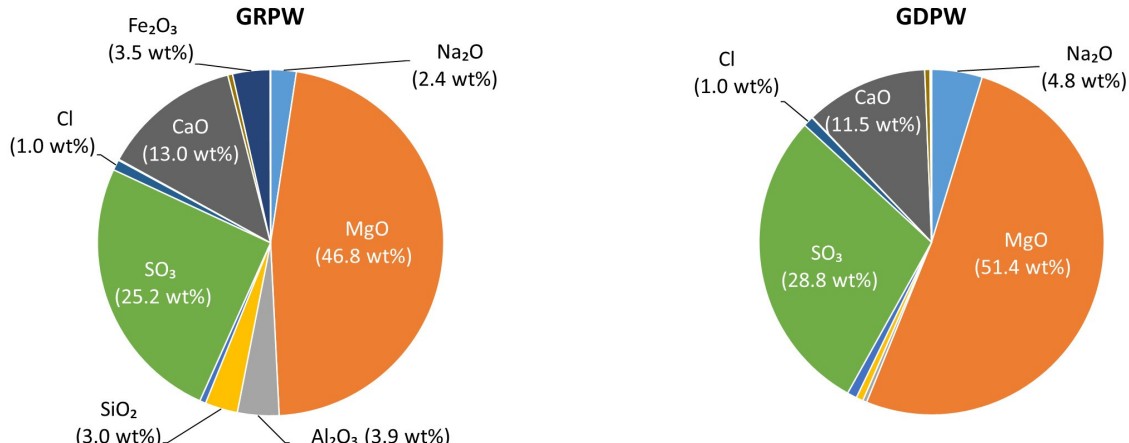

**Figure 4.** Composition of the precipitates produced at pH 10.5 from acidic wastewater obtained after acid leaching purification of gypsum from refurbishment plasterboard waste (GRPW) and gypsum from demolition plasterboard waste (GDPW) at 90 °C for 1 h using a 5 wt% $H_2SO_4$ solution prepared with municipal water and a stirring rate of 50 rpm. Only constituents with contents ≥1 wt% are displayed.

In order to determine the impact of treated water reuse, acid leaching purification of GDPW was carried out at 90 °C for 2 h using a 3 wt% $H_2SO_4$ solution prepared with purified water and a stirring rate of 50 rpm. Under these acidic leaching conditions, the

chemical purity of the resulting gypsum was above 96 wt%. However, these conditions are not optimum because the purity level after the purification of different batches might not be consistent, as suggested by the lower chemical purity level of the samples in our previous work [2]. The recycled GDPW was filtered, and the acidic wastewater was treated with $Ca(OH)_2$ to pH 10.5. The resulting precipitate was filtered, and the recovered treated water (treated water 1) was reused to prepare the 3 wt% $H_2SO_4$ solution for the next acid leaching purification test. This methodology was repeated once again to produce recycled GDPW using treated water 2. Figure 5 presents the chemical purity of recycled GDPW when using purified water, treated water 1, and treated water 2. The results show that there was no variation in the chemical purity of the samples within experimental error. The $CaSO_4$ content in recycled gypsum dropped by 1.3 wt% when treated water 1 was used. However, the $CaSO_4$ content in recycled gypsum when treated water 2 was used (94.6 wt%) was similar to that when purified water was used (94.3 wt%).

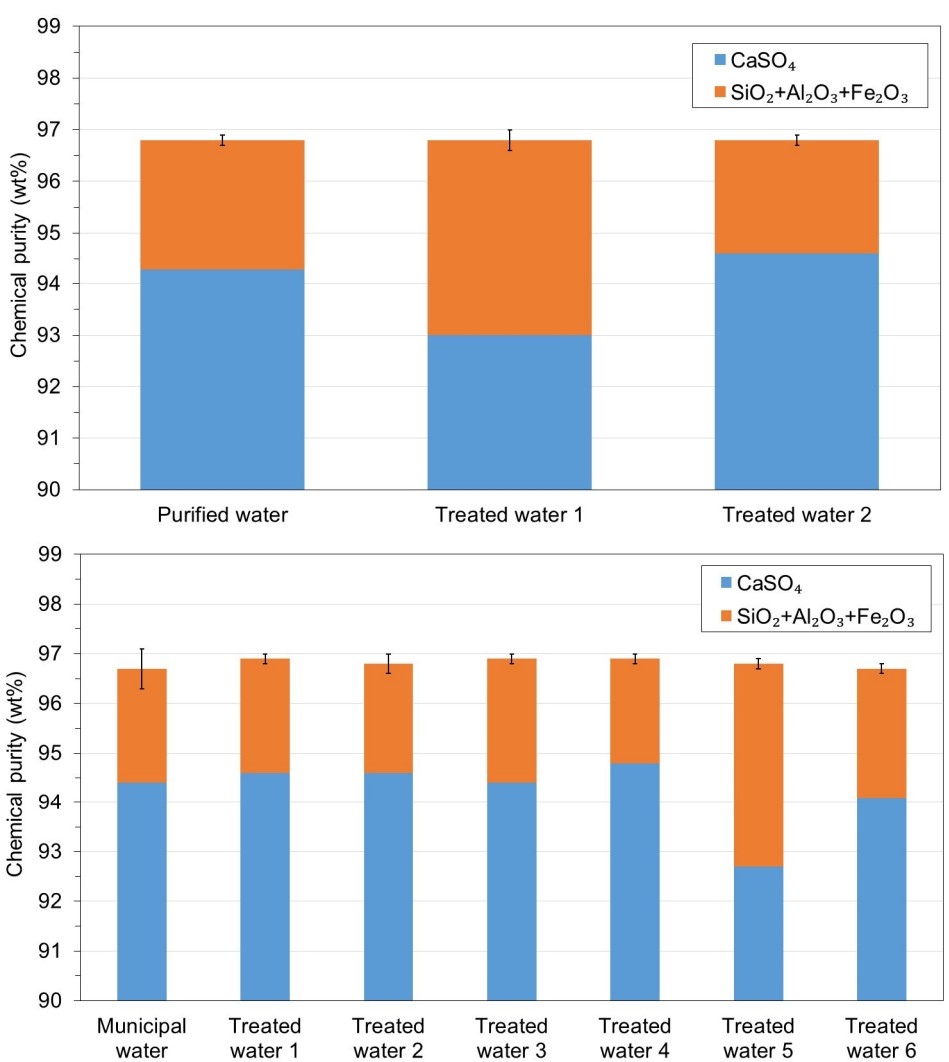

**Figure 5.** Chemical purity of GDPW (**top**) and GRPW (**bottom**) after acid leaching purification at 90 °C for 2 h using a 3 wt% $H_2SO_4$ solution prepared with purified water when using GDPW, municipal water when using GRPW, treated waters, and a stirring rate of 50 rpm.

The chemical composition of the precipitates from GDPW was determined through XRF (Figure S2 in Supplementary Materials). The MgO content in the precipitate increased from 18.0 wt% with purified water to 21.8 wt% with treated water 1 and to 26.2 wt% with treated water 2. In addition, the sum of $CaSO_4$ and MgO contents in the precipitates decreased from approximately 95 wt% when using purified water to 93 wt% when using

treated water 1 and to 91 wt% when using treated water 2. There was also a simultaneous increase in Fe and Si compounds in the precipitate. XRD results in Table 2 indicate that the gypsum content in the precipitates increased from 79.0 wt% with purified water to 81.7 wt% with treated water 1 and to 87.7 wt% with purified water 2. There were significant amounts of kieserite (6.1–9.6 wt%) and brucite (5.2–8.9 wt%) in the precipitates.

In further acid leaching gypsum purification tests, municipal water was used instead of purified water to determine its impact on process efficiency and precipitate composition. Acid leaching purification of GRPW was carried out at 90 °C for 2 h using a 3 wt% $H_2SO_4$ solution prepared with municipal water and a stirring rate of 50 rpm. Initially, the gypsum slurry was neutralized with $Ca(OH)_2$ to pH 5.5, and the resulting wastewater was treated with $Ca(OH)_2$ to pH 10.5 to recover the precipitate and treated water 1. Treated water 1 was used to prepare the 3 wt% $H_2SO_4$ solution of the second acid leaching purification test. As for GDPW, the acidic wastewater obtained in the second acid leaching purification test was treated with $Ca(OH)_2$ to pH 10.5. The resulting precipitate was filtered, and the recovered treated water (treated water 2) was reused to prepare the 3 wt% $H_2SO_4$ solution for the next acid leaching purification test. In total, six cycles with treated water reuse were performed. The chemical purity of the recycled GRPW samples obtained when using municipal water and reused treated waters was determined (Figure 5). The chemical purity of the samples was not affected by the reuse of the treated water, although the $CaSO_4$ content increased in the first four cycles. The low $CaSO_4$ content in the recycled GRPW when using treated water 5 could be due to the higher pH achieved in the gypsum slurry neutralization stage (pH 7.0 rather than pH 5.5). Nevertheless, the consistent chemical purity of the samples suggests that treated water obtained at pH 10.5 could be reused for at least six cycles without impacting the acid leaching gypsum purification process efficiency. In addition, the use of municipal water rather than purified water should not cause a reduction in the chemical purity of the recycled gypsum. Table S1 in the Supplementary Materials presents the chemical composition of the precipitates obtained at pH 10.5 when using municipal water or treated water obtained in six acid leaching cycles. The main oxides in the precipitates were MgO (>20 wt%), $SO_3$ (>20 wt%), and CaO (>12 wt%). $Na_2O$ and $Fe_2O_3$ were also present in significant amounts in most precipitates (typically between 3 and 11 wt%) and were higher than those found in the precipitates from GDPW (Figure S2 in Supplementary Materials). Table 1 shows that the $Na_2O$ content in GRPW (0.5 wt%) is higher than the $Na_2O$ content in GDPW (<0.1 wt%). On the other hand, the $Fe_2O_3$ content in GRPW and GDPW is similar (0.2–0.4 wt%). It must be noted that the acid leaching purification tests with GRPW were performed with municipal water, which may contain traces of Na and Fe salts. Therefore, the higher $Na_2O$ and $Fe_2O_3$ contents in the precipitates from GRPW could be due to the different contents of Na and Fe compounds in GRPW and GDPW and the possible presence of these compounds in municipal water. $SiO_2$, $Al_2O_3$, and $P_2O_5$ contents were ≤4 wt%, Cl content was between 1 and 2 wt%, and $K_2O$ and MnO contents were <1 wt%. The MgO content in the precipitate increased in the first two cycles, similarly to the precipitates obtained from GDPW (Figure S2 in Supplementary Materials). However, the MgO content in the precipitates from subsequent cycles did not follow a pattern.

### 3.3. WMA 3

The addition of $Ca(OH)_2$ to neutralize the spent $H_2SO_4$ solution prior to filtration would avoid the use of expensive corrosive-resistant pumps and filtration equipment. However, soluble impurities could precipitate during neutralization, thus reducing the chemical purity of the recycled gypsum. Two acid leaching purification tests were performed with GRPW at 90 °C for 1 h using 5 wt% $H_2SO_4$ solutions prepared with purified water. In the first acid leaching purification test, the purified gypsum slurry was filtered and washed with purified water as per WMA 2. In the second acid leaching purification test, the purified gypsum slurry was neutralized with $Ca(OH)_2$ to pH 5.5 and then filtered without washing. Neutralization only caused reductions of 0.3 wt% in chemical purity

and 0.8 wt% in $CaSO_4$ content compared to washing (Figure 6). Therefore, neutralization seems to be a cheaper alternative to washing (high water consumption and wastewater production) to preserve the chemical purity of the gypsum product.

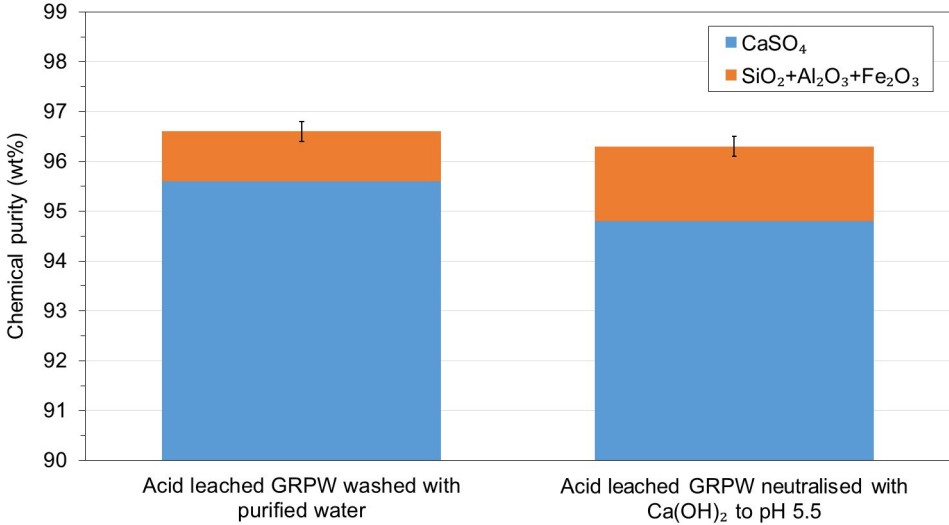

**Figure 6.** Comparison of the chemical purity of GRPW after acid leaching at 90 °C for 1 h using a 5 wt% $H_2SO_4$ solution prepared with purified water and a stirring rate of 50 rpm after washing the gypsum cake with purified water during filtration and after neutralization of the gypsum slurry with $Ca(OH)_2$ to pH 5.5 prior to filtration.

Acid leaching purification tests were then carried out in triplicate with GRPW at 90 °C for 2 h using 3 wt% $H_2SO_4$ solutions, followed by neutralization with $Ca(OH)_2$ to pH 5.5 to verify the reproducibility of the results (Figure 7).

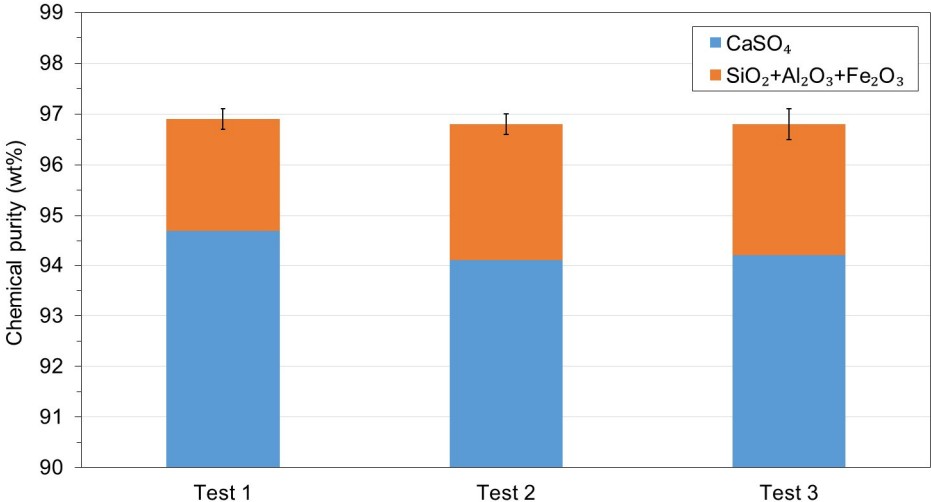

**Figure 7.** Chemical purity of GRPW after acid leaching at 90 °C for 2 h using 3 wt% $H_2SO_4$ solutions prepared with municipal water using a stirring rate of 50 rpm after neutralization with $Ca(OH)_2$ to pH 5.5 (triplicate tests).

The chemical purity of the gypsum samples showed good reproducibility (around 96.8 wt%). However, the $CaSO_4$ content varied between 94.1 wt% and 94.7 wt%. Overall, it could be argued that the acid leaching–neutralization–filtration–precipitation approach (WMA 3) could be a viable industrial-scale process configuration for acid leaching purification of gypsum from post-consumer plasterboard wastes.

*3.4. LCA*

Table 3 presents the LCA characterization analysis encompassing eighteen impact categories for the three sustainable wastewater management approaches. The results show that WMA 3 has the lowest $CO_2$e emissions and WMA 2 has the highest $CO_2$e emissions in the global warming impact category. Across the 18 impact categories, WMA 3 exhibits the lowest environmental impact in 12 categories, including global warming, stratospheric ozone depletion, ozone formation (human health), fine particulate matter formation, ozone formation (terrestrial ecosystems), terrestrial acidification, freshwater eutrophication, marine eutrophication, terrestrial ecotoxicity, human carcinogenic toxicity, human non-carcinogenic toxicity, and mineral resource scarcity. On the other hand, WMA 1 shows the lowest environmental impact in three categories, including ionizing radiation, land use, and water consumption. WMA 1 and WMA 3 exhibit the lowest environmental impact in the remaining categories, including freshwater ecotoxicity, marine ecotoxicity, and fossil resource scarcity.

**Table 3.** LCA characterization analysis for each sustainable wastewater management approach (WMA).

| Impact Category | Unit | WMA 1 | WMA 2 | WMA 3 |
|---|---|---|---|---|
| Global warming | kg $CO_2$ eq | 1551 | 1762 | 1545 |
| Stratospheric ozone depletion | kg CFC11 eq | 0.000595 | 0.000696 | 0.000564 |
| Ionizing radiation | kBq Co-60 eq | 396.60 | 544.52 | 410.41 |
| Ozone formation (human health) | kg $NO_x$ eq | 2.962 | 3.310 | 2.788 |
| Fine particulate matter formation | kg $PM_{2.5}$ eq | 3.725 | 3.851 | 3.514 |
| Ozone formation (terrestrial ecosystems) | kg $NO_x$ eq | 3.021 | 3.372 | 2.843 |
| Terrestrial acidification | kg $SO_2$ eq | 11.93 | 12.31 | 11.78 |
| Freshwater eutrophication | kg P eq | 0.689 | 0.684 | 0.645 |
| Marine eutrophication | kg N eq | 1.447 | 1.349 | 1.333 |
| Terrestrial ecotoxicity | kg 1,4-DCB | 19,275 | 19,513 | 19,044 |
| Freshwater ecotoxicity | kg 1,4-DCB | 186.4 | 191.3 | 185.4 |
| Marine ecotoxicity | kg 1,4-DCB | 244.8 | 251.5 | 243.3 |
| Human carcinogenic toxicity | kg 1,4-DCB | 101.7 | 101.5 | 93.9 |
| Human non-carcinogenic toxicity | kg 1,4-DCB | 3652 | 3713 | 3587 |
| Land use | $m^2$a crop eq | 83.6 | 103.9 | 86.1 |
| Mineral resource scarcity | kg Cu eq | −0.607 | −0.466 | −6.230 |
| Fossil resource scarcity | kg oil eq | 461.7 | 539.2 | 461.8 |
| Water consumption | $m^3$ | 38.64 | 40.63 | 41.62 |

Table S2 in the Supplementary Materials illustrates the LCA normalization analysis, thus enabling a comparable assessment of all impact categories within the life cycle assessment method. Human carcinogenic toxicity has the highest impact among the 18 impact categories in all sustainable wastewater management approaches. Freshwater ecotoxicity has the second highest overall impact, while marine ecotoxicity ranks third and terrestrial ecotoxicity ranks fourth. Overall, the results demonstrate that WMA 3 exhibits the best environmental performance and WMA 2 exhibits the worst environmental performance. Therefore, the steps of the recommended in-house wastewater treatment would be acid leaching, acid neutralization, purified gypsum filtration, purified gypsum cake drying, precipitation of soluble impurities in wastewater, precipitate filtration, precipitate drying, and reuse of treated water.

*3.5. Comparison of WMA 3 with Other Acidic Wastewater Technologies*

Several stepwise processes have been proposed to treat acidic wastewater. A three-step process was proposed by Zhang et al. [32] to treat highly acidic wastewater derived from $TiO_2$ production. In the first step, high-quality gypsum was produced after neutralization with $CaCO_3$ to pH 2. In the second step, schwertmannite ($Fe_8O_8(OH)_{8-2x}(SO_4)_x$, x = 1–1.75) was formed through the reaction of $FeSO_4$ with $H_2O_2$ with stirring for 24 h. In the third



step, a NaOH solution was added to adjust the pH to 7.5, which resulted in the precipitation of metals as hydroxides after stirring for 24 h. In another study, Salo et al. [33] applied biological sulfate reduction to leachates obtained after acid leaching of phosphogypsum. This biological treatment produced a precipitate concentrating the rare earth elements present in phosphogypsum and converted $SO_4^{2-}$ into $S^{2-}$ in the liquid phase. However, the efficiency of the bioreactor was highly dependent on the acidity of the wastewater, which would limit process control and scalability, and a hydraulic retention time of 38 h was required under optimum bioreactor conditions. Xiong et al. [30] also developed a laboratory-scale process consisting of precipitation, acid leaching, and oxidation steps to recover $Mg(OH)_2$ from a leachate of dolomitic phosphate ore. The leachate liquor was neutralized with $Ca(OH)_2$ to pH 7 to precipitate $Fe(OH)_3$. Then, filtered leachate was neutralized with $Ca(OH)_2$ to pH 10.0 to precipitate $Mg(OH)_2$. The $Mg(OH)_2$ product was further purified through acid leaching, oxidation, and precipitation steps. However, numerous chemicals ($H_2SO_4$, $NH_4OH$, $Mg(OH)_2$, $(NH_4)_2S_2O_8$, and $NH_3$) were needed to achieve high $Mg(OH)_2$ recovery yields, which would significantly increase material costs. In comparison, only $H_2SO_4$ and $Ca(OH)_2$ were required in WMA 3 (Figure 8). Therefore, the novelty of the in-house wastewater treatment based on WMA 3 proposed in this work is the minimum economic impact on the acid leaching gypsum purification plant whilst preserving the high purity (>96 wt%) of the gypsum product, the valorization of the Mg-rich gypsum by-product, and the reduction of the environmental impact by reusing the treated water. Material costs could be reduced further by replacing commercial $Ca(OH)_2$ with by-products from quicklime manufacturing [34]. The reuse of the treated water would adhere to the EU's new Circular Economy Action Plan [7], which promotes water reuse and efficiency in industrial processes.

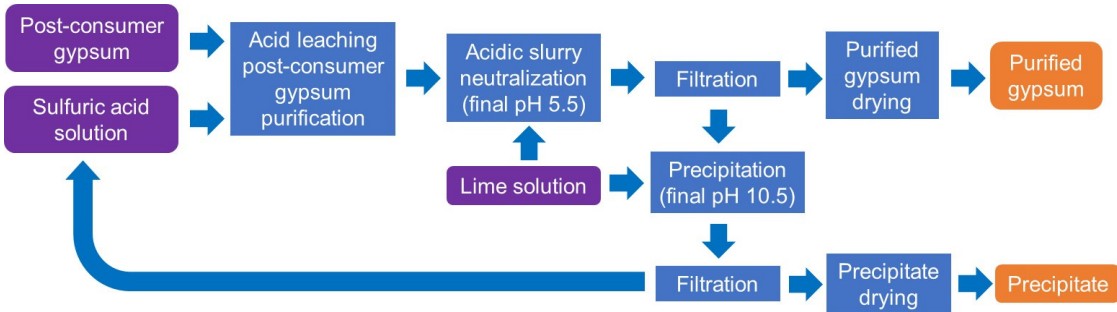

**Figure 8.** Roadmap of sustainable wastewater management approach 3.

### 3.6. By-Product Applications

The EU's Directive 2008/98/EC31 classifies the acidic wastewater obtained after acid leaching purification of gypsum from consumer plasterboard wastes as a hazardous by-product because it is highly corrosive (pH < 1). Therefore, acidic wastewater must be neutralized to around pH 6 before it can be considered for commercial application. The neutralized wastewater, which would mainly contain magnesium and calcium sulfates, could be used as a liquid fertilizer, but there are two major issues related to its commercial exploitation. Firstly, the gypsum/solution ratio of 1:3 wt/wt used during acid leaching would imply that around 3000 L of liquid fertilizer would be produced per ton of recycled gypsum. As a result, the water consumption in the acid leaching process would be extremely high, which would lead to high operating costs at the acid leaching purification plant. Secondly, this liquid fertilizer might not have enough demand from the agricultural industry because of the high volumes produced.

On the other hand, the precipitation of soluble impurities in wastewater at pH 10.5 not only offers the advantage of producing much lower amounts of solid fertilizer but also treated water that can be reused without impacting the acid leaching gypsum purification process. The main disadvantages of precipitating the soluble impurities in the acidic wastewater at pH 10.5 are that approximately 30 wt% extra $Ca(OH)_2$ would be needed

compared to acidic wastewater neutralization to pH 5–6, and additional equipment, such as precipitation and storage tanks, a filter press, and pumps, would be required. As shown in Table 2, the precipitate obtained at pH 10.5 is a Mg-rich gypsum comprising 79.0–87.7 wt% $CaSO_4 \cdot 2H_2O$, 6.1–9.6 wt% $MgSO_4 \cdot H_2O$, 5.2–8.9 wt% $Mg(OH)_2$, and 1.0–4.0 wt% $Ca(OH)_2$. The Mg-rich gypsum can be classified as an inorganic macronutrient fertilizer [35,36] as it contains more than 1.5 wt% MgO, more than 1.5 wt% CaO, and more than 1.5 wt% $SO_3$. Furthermore, Ca and Mg compounds in the precipitate are considered secondary nutrient fertilizers [37]. Ritchey and Snuffer [38] indicated that abandoned pasture soils are particularly likely to be low in Ca and Mg species. These authors used gypsum supplemented with 5–6 wt% $Mg(OH)_2$ to maintain adequate Mg levels in the soil of an abandoned Appalachian pasture. In another study, Ayanda et al. [39] found that a Mg-rich gypsum with pH 8.8 that contained 45 wt% $CaSO_4 \cdot 2H_2O$, 17.1 wt% $Mg(OH)_2$, 4.3 wt% $Ca(OH)_2$, and 2.3 wt% $CaCO_3$ was an effective source of Ca and Mg for oil palm growth and a good soil ameliorant. Oil palm is one of the world's most important oil crops because it can produce more vegetable oil per unit of land than any other crop (e.g., soybean, rapeseed, sunflower), and it is currently being used as biofuel and as an ingredient in many processed foods, cosmetics, pharmaceuticals, etc. [40]. The global area utilized for oil palm growth increased from less than 4 million hectares in 1980 to 20 million hectares in 2018, and future global demand for palm oil is expected to increase [40]. The chemical composition of the Mg-rich gypsum used by Ayanda et al. [39] is comparable to that of the precipitate recovered from the acidic wastewater after acid leaching gypsum purification. In addition, a mass balance of the laboratory-scale acid leaching purification process indicated that around 30 kg of precipitate would be produced per ton of gypsum waste processed, which is 100 times lower than the amount of liquid fertilizer that would be produced after neutralization of the acidic wastewater. Therefore, the commercialization of the Mg-rich gypsum as fertilizer for oil palm soils and reuse of the treated water in the acid leaching process are proposed as the most compelling sustainable solutions to preserve recycled gypsum quality and minimize waste production and disposal costs. An alternative commercial use for the alkaline Mg-rich gypsum obtained at pH 10.5 could be acidic soil ameliorant.

### 3.7. Potential Barriers and Enabling Measures

Legal, social, technical, economic, and environmental barriers and enabling measures for the commercial implementation of the in-house wastewater treatment in the acid leaching purification plant for gypsum from post-consumer plasterboard wastes are presented in Table 4. These barriers and enabling measures were identified based on the work of Hukari et al. [41] and Dutta et al. [42]. It is envisaged that the most important barriers for the implementation of the in-house wastewater treatment will be economic, including, specifically, the additional equipment, materials, energy, and labor costs for the production of the Mg-rich gypsum (B4), and the limited demand for the Mg-rich gypsum as a fertilizer or soil ameliorant (B5).

**Table 4.** Potential barriers and enabling measures for the implementation of WMA 3 in an industrial acid leaching gypsum purification plant.

| Barriers (B) | | Enabling Measures (M) |
|---|---|---|
| Legal | **B1**. Lack of local, regional, national, and EU-wide permits and authorization processes for the installation and operation of acidic wastewater treatment plants and disposal of the treated water after maximum utilization cycles.<br><br>**B2**. Lack of regulations for the magnesium-rich gypsum as a fertilizer or soil ameliorant product. | **M1**. New local, national, and EU-wide regulations for acidic wastewater treatment plants and effluent disposal, or adaptation of existing regulations (e.g., the EU's Environmental Impact Assessment Directive [43]).<br>**M2**. Adaptation of End-of-Waste criteria of the EU's Waste Framework Directive [4], Fertilizers Regulation [35], and Registration, Evaluation, Authorization and Restriction of Chemicals (REACH) Regulation [44]. |

**Table 4.** *Cont.*

| Barriers (B) | | Enabling Measures (M) |
|---|---|---|
| Social | **B3**. Low acceptance of the magnesium-rich gypsum fertilizer or soil ameliorant by the agricultural industry due to lack of knowledge. | **M3**. Dissemination campaigns for practitioners (e.g., specialist magazines, workshops, etc.) to highlight sustainability benefits and provide guidance for applications as fertilizer or soil ameliorant. Wastewater sustainability guidelines issued by governments. |
| Technical | **B4**. Inconsistent quality of the magnesium-rich gypsum fertilizer or soil ameliorant, which restricts its commercialization. | **M4**. Quality control of incoming plasterboard waste and magnesium-rich gypsum by-product with training of operatives for effective wastewater treatment process management. |
| Economic | **B5**. Additional equipment, materials, energy, and labor costs in the acid leaching gypsum purification plant for the in-house wastewater treatment. | **M5**. Government incentives in the form of tax rebates and financial and technical assistance to plasterboard recyclers when implementing the wastewater treatment technology and commercializing the magnesium-rich gypsum as a fertilizer or soil ameliorant. |
| | **B6**. Non-existent market or limited demand for the magnesium-rich gypsum as a fertilizer or soil ameliorant. | **M6**. Identifying and targeting niche agricultural markets with high demand for the magnesium-rich gypsum by-product. |
| Environmental | **B7**. Adverse environmental impact on aquatic ecosystems after application of the magnesium-rich gypsum (e.g., water eutrophication). | **M7**. Research and development to monitor the magnesium-rich gypsum's mobility in soils and aquatic ecosystems. |

## 4. Conclusions

Three sustainable wastewater management approaches were evaluated in this work to minimize the economic and environmental impacts of an in-house wastewater treatment in the acid leaching purification plant for gypsum from post-consumer plasterboard wastes. These approaches mainly consisted of: (i) reusing the acidic wastewater obtained after gypsum purification, filtration, and washing; (ii) treating the acidic wastewater obtained after gypsum purification, filtration, and washing with precipitation of soluble impurities and reuse of the treated water; and (iii) neutralizing the acidic wastewater prior to filtration to reduce water consumption and avoid expensive, corrosion-resistant equipment, followed by precipitation and reuse of the treated water. The main findings of this work were:

1.  The reuse of acidic wastewater was not technically viable because there was no improvement in purified gypsum quality compared to the gypsum feedstock.
2.  A pH of 10.5 was required to precipitate $Mg(OH)_2$, and the precipitate was a Mg-rich gypsum mainly composed of $CaO$, $SO_3$, and $MgO$ ($\geq 85\%$ on a weight basis).
3.  The reuse of the treated water obtained after precipitation of the soluble impurities did not affect the chemical purity of the recycled gypsum after six cycles, thus enabling the reduction of water usage and wastewater disposal costs in the acid leaching gypsum purification plant.
4.  Acid neutralization prior to filtration did not reduce the chemical purity of the recycled gypsum but decreased its $CaSO_4$ content by 0.8 wt%. The economic and environmental benefits of avoiding recycled gypsum cake washing and expensive, corrosion-resistant equipment at the acid leaching gypsum purification plant would greatly compensate for this small reduction in $CaSO_4$ content.
5.  The steps of the proposed in-house wastewater treatment are acid leaching, acid neutralization, purified gypsum (chemical purity > 96 wt%) filtration, purified gypsum cake drying, precipitation of soluble impurities in wastewater (Mg-rich gypsum), precipitate filtration, precipitate drying, and reuse of treated water in the acid leaching step.

The novelty of this work lies in the development of an in-house wastewater treatment for an acid leaching gypsum purification plant that exhibits the lowest environmental impact and minimizes the economic impact. The in-house wastewater treatment will enable the reuse of the treated water in the acid leaching gypsum purification process and the recovery and exploitation of a Mg-rich gypsum as fertilizer or soil ameliorant for agricultural applications. The implementation of the in-house wastewater treatment in the acid leaching gypsum purification plant could potentially be restricted by legal, social, economic, technical, and environmental barriers, with economic barriers being the most important due to the additional equipment, material, and labor required and the foreseen limited demand for the Mg-rich gypsum by-product for agricultural applications. Future research will focus on technical, environmental, and economic evaluations of an industrial-scale acid leaching gypsum purification plant with the capacity to process 2 tons of gypsum waste per day.

**Supplementary Materials:** The following supporting information can be downloaded at: https://www.mdpi.com/article/10.3390/su16010425/s1, Table S1: Composition of the precipitates obtained at pH 10.5 from acidic wastewater obtained after sequential acid leaching tests of GRPW at 90 °C for 2 h using a 3 wt% $H_2SO_4$ solution and a stirring rate of 50 rpm; Table S2: LCA normalization analysis for the three approaches evaluated in this work; Figure S1: Beaker with wastewater at pH 5.5 (a), pH 8.0 (b), and pH 10.5 (c); Figure S2: Chemical composition of precipitates produced at pH 10.5 from acidic wastewater obtained after acid leaching of GDPW at 90 °C for 2 h using 3 wt% $H_2SO_4$ solutions prepared with purified water (top), treated water 1 (middle), and treated water 2 (bottom) and a stirring rate of 50 rpm.

**Author Contributions:** Conceptualization, S.C. and M.O.; methodology, M.C.-D., S.C. and M.O.; validation, P.N., B.P. and T.L.; formal analysis, M.C.-D., S.M. and M.G.; investigation, M.C.-D.; resources, P.N., B.P. and T.L.; data curation, M.C.-D. and S.M.; writing—original draft preparation, M.C.-D.; writing—review and editing, S.C., M.O., S.M., M.G., P.N., B.P. and T.L.; supervision, S.C. and M.O.; project administration, S.C. and M.O.; funding acquisition, S.C. and M.O. All authors have read and agreed to the published version of the manuscript.

**Funding:** This project has received funding from the European Union's Horizon 2020 research and innovation program under grant agreement No 869336.

**Institutional Review Board Statement:** Not applicable.

**Informed Consent Statement:** Not applicable.

**Data Availability Statement:** XRF and XRD spectra supporting the reported results in this article can be found in Zenodo (https://doi.org/10.5281/zenodo.10397236).

**Acknowledgments:** The authors would like to acknowledge the use of XRF and XRD facilities within the Loughborough Materials Characterization Centre (LMCC) at Loughborough University.

**Conflicts of Interest:** ENVA is a leading provider of recycling and resource recovery solutions for re-use in manufacturing and energy conversion. British Gypsum, which is part of the Saint-Gobain Group, is a leading manufacturer of plasterboard and plaster-based drylining systems and products. ENVA have vested interests in the outcomes of this research and future commercialization of the wastewater treatment and by-products developed in this paper. British Gypsum do not have vested interests in the exploitation of the wastewater treatment developed in this paper.

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
