# Peer review of "Valorization of Acid Leaching Post-Consumer Gypsum Purification Wastewater"

_sustainability, doi:10.3390/su16010425_

Round 1

Reviewer 1 Report

Comments and Suggestions for Authors

The authors present a topic of revaluation of process byproducts rich in minerals.

The work contains valuable information, however the authors must present the XRF images or micrographs that they used for quantification. Also, did they use any other analytical methods? Considering that XRF is a superficial analysis technique.

The XRD must be presented in the manuscript, they are mentioned in the methodology, but there are no results.

Author Response

1. The work contains valuable information, however the authors must present the XRF images or micrographs that they used for quantification. Also, did they use any other analytical methods? Considering that XRF is a superficial analysis technique.

XRF spectra have been uploaded in Zenodo (https://doi.org/10.5281/zenodo.10397236) and this has been added to the Data Availability Statement of the article. No other analytical techniques have been used apart from the ones mentioned in the manuscript (XRF and XRD).

2. The XRD must be presented in the manuscript, they are mentioned in the methodology, but there are no results.

The XRD results are presented in Table 2. The XRD spectra have been uploaded in Zenodo (https://doi.org/10.5281/zenodo.10397236) and this has been added to the Data Availability Statement of the article.

Reviewer 2 Report

Comments and Suggestions for Authors

This paper focused on the recovery of valuable solid product and reusable water from highly acidic wastewater. The works were enrich and wonderful results were obtained. However, some issues should be concerned:

1. Where was the XRD pattern of MgO?

2. How to obtain the results showed in Table 2? According to what?

3. What were the treated water 1-5 in Figure 5?

4. The figures needed to improve.

Author Response

1. Where was the XRD pattern of MgO?

The MgO peak is determined through XRF and not XRD. The MgO peak was determined by fitting the XRF spectra to standard peaks of the different oxides stored in the XRF software library and it is positioned at 1.2 keV in the spectra. The Mg minerals determined through XRD are also determined by matching the spectra with standard spectra in the XRD software library.

2. How to obtain the results showed in Table 2? According to what?

The results presented in Table 2 were obtained by processing the XRD spectra. A sentence has been introduced in the Methods section of the manuscript to clarify this: “The XRD patterns were analyzed with DIFFRAC.EVA diffraction software”.

3. What were the treated water 1-5 in Figure 5?

This has been explained in the manuscript. The treated water 1-5 come from the processing of the wastewater obtained in the previous test. For instance, treated water 3 is obtained after processing the wastewater generated after using treated water 2 in the acid leaching post-consumer gypsum purification process. A sentence in the Results section has been introduced to clarify this.

4. The figures needed to improve.

The chemical formulas in all figures (Manuscript + Supplementary Materials) have been corrected.

Reviewer 3 Report

Comments and Suggestions for Authors

The work explores the treatment of highly acidic wastewater generated during the purification of gypsum from post-consumer plasterboard wastes. The study aims to recover valuable solid products and reusable water while minimizing the economic and environmental impacts of in-house wastewater treatment. Through the evaluation of different wastewater management approaches, the research seeks to enhance the efficiency of the purification process and the valorization of impurities in the wastewater. Additionally, the environmental performance of the treatment methods is assessed using the life cycle assessment (LCA) method, providing insights into the sustainability implications of the proposed approaches. Some comments are listed below:

(1) The format of the Figures should be further modified, such as the writing of chemical formulas.

(2) XRF is not a good technique to determine the multi-element content of samples because only semi-quantitative results can be obtained. It would be better to supplement the quantitative multi-element analysis results of samples.

(3) It would be better to provide the XRD pattern of samples before and after treatment if you measured.

(4) The paper should include a roadmap for the treatment of highly acidic wastewater generated during the purification of gypsum from post-consumer plasterboard wastes.

(5) Conclusion part should be more concise.

Author Response

1. The format of the Figures should be further modified, such as the writing of chemical formulas.

The chemical formulas in all figures have been corrected.

2. XRF is not a good technique to determine the multi-element content of samples because only semi-quantitative results can be obtained. It would be better to supplement the quantitative multi-element analysis results of samples.

The authors could not use additional analytical techniques to characterize the materials. Furthermore, past research studies have used XRF to determine the chemical composition of gypsum products such as phosphogypsum.

3. It would be better to provide the XRD pattern of samples before and after treatment if you measured.

The XRD patterns have been uploaded in Zenodo (https://doi.org/10.5281/zenodo.10397236) and this has been added to the Data Availability Statement of the article.

4. The paper should include a roadmap for the treatment of highly acidic wastewater generated during the purification of gypsum from post-consumer plasterboard wastes.

The roadmap of the proposed wastewater management approach 3 has been included in section 3.5 (Figure 8).

5. Conclusion part should be more concise.

The Conclusions section summarizes the most important findings of this research work, highlights their implications for stakeholders and recommends next steps in this research area. Therefore, the authors have not modified the Conclusions section of the manuscript.

Round 2

Reviewer 1 Report

Comments and Suggestions for Authors

The authors improved their work in some points that were observed.

Reviewer 3 Report

Comments and Suggestions for Authors

The authors have addressed my concerns well. The paper can be accepted in the current version.